

# Effects of mechanical and chemical control on invasive *Spartina alterniflora* in the Yellow River Delta, China

Baohua Xie[1], Guangxuan Han[1], Peiyang Qiao[1,2], Baoling Mei[2], Qing Wang[1], Yingfeng Zhou[3], Anfeng Zhang[3], Weimin Song[1] and Bo Guan[1]

[1] Key Laboratory of Coastal Zone Environmental Processes and Ecological Remediation, CAS, Shandong Provincial Key Laboratory of Coastal Environmental Processes, Yantai Institute of Coastal Zone Research, Chinese Academy of Sciences, Yantai, China
[2] College of Ecology and Environment, Inner Mongolia University, Hohhot, China
[3] Administration Bureau of the Yellow River Delta National Nature Reserve, Dongying, China

## ABSTRACT

*Spartina alterniflora* is one of the most noxious invasive plants in China and many other regions. Exploring environmentally friendly, economic and effective techniques for controlling *Spartina alterniflora* is of great significance for the management of coastal wetlands. In the present study, different approaches, including mowing and waterlogging, mowing and tilling and herbicide application, were used to control *Spartina alterniflora*. The results suggest that the integrated approach of mowing and waterlogging could eradicate *Spartina alterniflora*, the herbicide haloxyfop-r-methyl could kill almost all the *Spartina alterniflora*, and the integrated approach of mowing and tilling at the end of the growing season was a perfect way to inhibit the germination of *Spartina alterniflora* in the following year. However, no matter which control approach is adopted, secondary invasion of *Spartina alterniflora* must be avoided. Otherwise, all the efforts will be wasted in a few years.

# INTRODUCTION

*Spartina alterniflora* Loisel (smooth cordgrass) is a perennial $C_4$ grass native to the eastern and gulf coasts of the United States and has important ecological functions in its native ecosystems (*Mobberley, 1956*). Due to intentional or unintentional introduction, *Spartina alterniflora* is now distributed in coastal marshes almost all around the world. Because of its vigorous vitality, strong salt tolerance, waterlogging tolerance, strong asexual reproduction and rapid expansion, *Spartina alterniflora* poses a serious threat to the biodiversity and ecological security of many coastal tidal wetlands (*Li et al., 2009*; *Strong & Ayres, 2013*). It is now a notorious invader of coastal ecosystems in many regions of the world, including estuaries in New Zealand, China, Africa and the Pacific coast of the USA (*Adams, van Wyk & Riddin, 2016*; *An et al., 2007*; *Buhle, Feist & Hilborn, 2012*; *Knott, Webster & Nabukalu, 2013*).

Spartina alterniflora was artificially introduced to China in 1979 (*An et al., 2007*) and can now be found in all coastal provinces of the country (*Liu et al., 2018*). Since it was

Corresponding author
Guangxuan Han, gxhan@yic.ac.cn

artificially introduced to the Yellow River Delta (YRD) in 1990, *S. alterniflora* has become widely distributed in the intertidal zone of the YRD, with a total area of 3,278 ha in 2015 (*Yang et al., 2017*). In the invaded area in the YRD, zooplankton biomass and diversity have decreased, benthic species abundance has decreased, economic shellfish have disappeared, and bird foraging and habitat have also become threatened (*Shen et al., 2009*; *Tian et al., 2009*; *Tian et al., 2008*). *Spartina alterniflora* has also significantly altered the soil physicochemical characteristics and microbial communities in the YRD (*Zhang et al., 2018*).

In order to minimize its negative ecological effects, the control of *Spartina alterniflora* has become an important issue in coastal wetland management. *Spartina alterniflora* is a perennial herb, and its reproductive modes include sexual reproduction via seeds and asexual reproduction via rhizomes or plant fragments (*Strong & Ayres, 2016*). The objectives of the various approaches used to control *Spartina alterniflora* are to solely or simultaneously inhibit its growth, sexual reproduction and asexual reproduction. Managers and scientists have attempted to develop techniques for controlling this species, including mechanical, chemical and biological approaches (*Knott, Webster & Nabukalu, 2013*; *Gao et al., 2014*; *Xie et al., 2018*). Some mechanical approaches, such as mowing and flooding, mowing and shading, can achieve good weeding effect and have little impact on the environment (*Yuan et al., 2011*; *Smith & Lee, 2015*). Chemical control methods usually use herbicides, which are easy to implement and have achieved good control effect in some areas. However, herbicides may cause environmental pollution and damage the health of animals and plants (*Patten, O'Casey & Metzger, 2017*; *Qiao et al., 2019*). Biological control methods need to be improved, and there is also a risk of ecological invasion of new alien species (*Xie et al., 2018*).

Although some of the previous approaches can achieve good control results, there are still many aspects to be improved: (1) The control efficacy of an approach may vary greatly in different regions (*Patten, 2004*); (2) the cost of control needs to be further reduced (*Riddin, van Wyk & Adams, 2016*; *Yuan et al., 2011*); (3) it takes too long to eliminate or eradicate *Spartina alterniflora*, ranging from a few years to more than a decade (*Kerr et al., 2016*; *Patten, 2004*; *Patten, O'Casey & Metzger, 2017*; *Riddin, van Wyk & Adams, 2016*). The drawbacks of the control approaches may be due to the various growth periods of *Spartina alterniflora* or environmental conditions, such as climate, topography and elevation.

A series of in situ field experiments which included mechanical and chemical control methods were performed in the YRD of China during 2016–2018. The aim of this study is to explore or improve the control technology of *Spartina alterniflora* so as to reduce its cost, improve its efficiency and widen its application.

# MATERIALS AND METHODS

## Site description

The YRD (118°20′E–119°20′E, 37°16′N–38°16′N), one of the most active regions of land-ocean interaction in the world, is a fan-shaped area located on the southern bank of the Bohai Sea and the western part of Laizhou Bay in China. The YRD has a

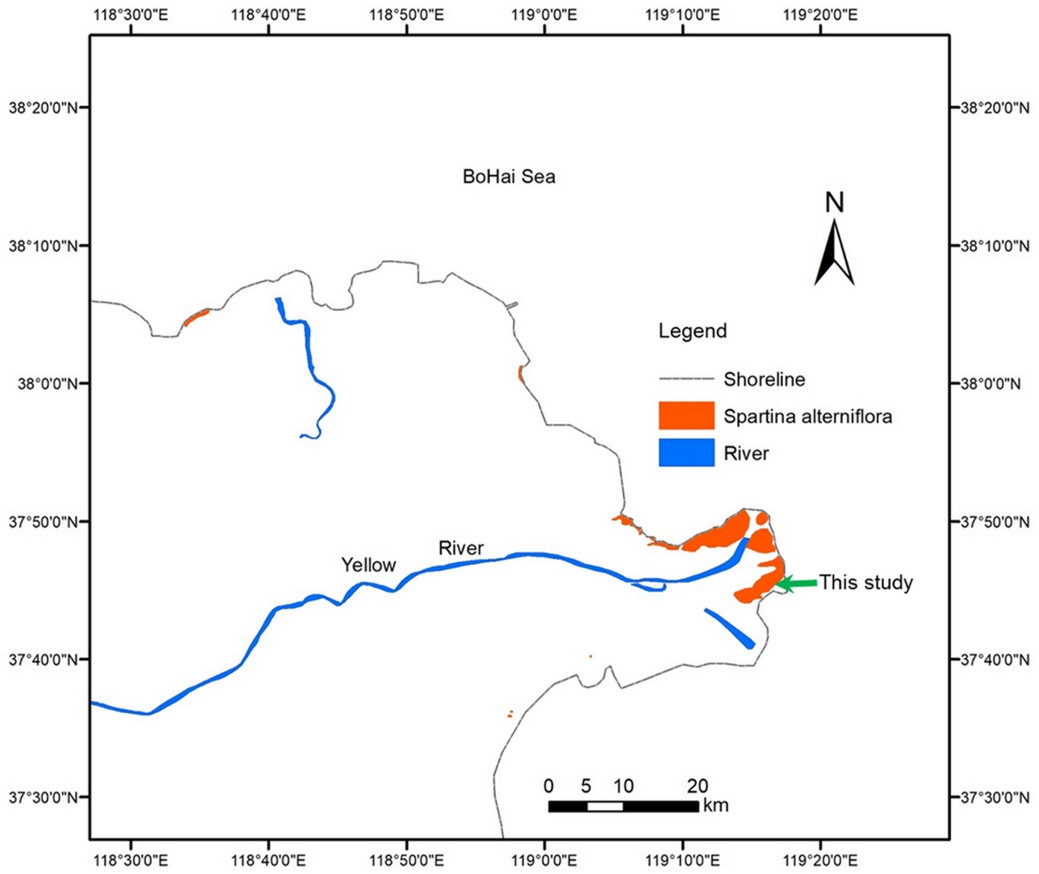

**Figure 1 Distribution of *Spartina alterniflora* in the Yellow River Delta of China.**

warm-temperate and semi-humid continental monsoon climate with distinctive seasons and a rainy summer. The annual average temperature is 12.9 °C, with minimum and maximum mean daily temperatures of −2.8 °C in January and 26.7 °C in July, respectively. The average annual precipitation is 560 mm, nearly 70% of which occurs from July to September (*Han et al., 2018*). The in situ field experiment site in this study is located on the south side of the Yellow River estuary (37°43′46.36″N, 119°15′13.29″E). The area has frequent tides, and the highest tide level exceeds two m. *Zostera japonica, Spartina alterniflora, Suaeda salsa, Phragmites communis* and *Tamarix chinensis* are sequentially distributed from sea to land. The niche of *Spartina alterniflora* overlaps with that of *Z. japonica* and *Suaeda salsa* on the low-tidal and mid-tidal beaches, respectively. Figure 1 shows the experimental site of this study and the distribution of *Spartina alterniflora* in YRD.

## Experimental design
### Mowing and waterlogging to control *Spartina alterniflora*
*Control of the clonal ramets of* Spartina alterniflora
The reproductive modes of *Spartina alterniflora* include sexual reproduction via seeds and asexual reproduction via rhizomes. Accordingly, the sprouts of *Spartina alterniflora*
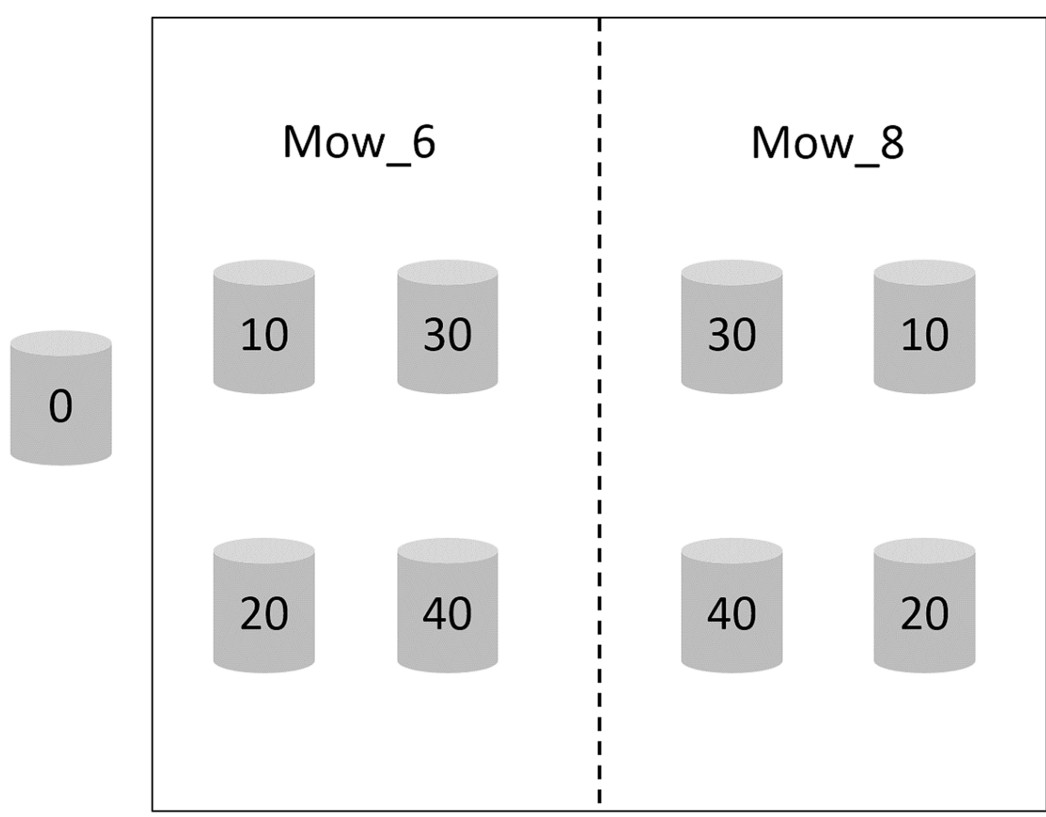

**Figure 2 Schematic diagram of the integrated control experiment involving mowing and waterlogging.** Mow_6 and Mow_8 indicate that the time of mowing was early June or early August, respectively. Zero, 10, 20, 30 and 40 indicate the depth of waterlogging (cm).

include seedlings and clonal ramets. To explore the optimal combination scheme of mowing and waterlogging for the control of clonal ramets, experiments involving the interaction between mowing and waterlogging were designed.

Two mowing treatments, which were mowing in early June and early August 2017 (hereafter referred to as Mow_6 and Mow_8), were established, each having six replicates. The height of the stubble was less than two cm. There were five waterlogging treatments in each mowing plot, and the waterlogging levels were 0, 10, 20, 30 and 40 cm, respectively. Waterlogging lasted from early June 2017 to November 2018. The zero cm waterlogging treatment is considered as the control treatment (CK) and was neither mowed nor waterlogged. The area of each mowing plot was approximately 10 m$^2$, in which PVC pipes with an inner diameter of 31 cm were buried to 40 cm underground. The heights of the PVC pipes above the ground were 0, 10, 20, 30 and 40 cm, respectively (Fig. 2). Tidal water was trapped in the pipes and maintained at the corresponding water level.

*Effects of different stubble heights on clonal ramet control efficacy*
Under a given waterlogging level, a lower stubble height results in a better control effect. When a mechanical equipment is used to mow *Spartina alterniflora*, the height of the stubble is unlikely to be zero cm. However, the height of the stubble can be easily limited to

less than 10 cm or even five cm. An experiment testing the interaction between stubble height and waterlogging level was established in early June 2018. The height of stubble was five or 10 cm. The depth of waterlogging was 0, 30 or 40 cm. According to the experimental results from 2017, the treatment of five cm stubble did not include 40 cm of waterlogging.

*Control of the seedlings of* Spartina alterniflora

The seedlings are slender and grow slowly in the early stages of growth. Although previous studies have shown that the integration of mowing and waterlogging can kill seedlings, this mortality may be caused by trampling during mowing. To determine the effect of waterlogging on the growth of seedlings, eight waterlogging plots were constructed in late May 2017 using PVC pipes with an inner diameter of 31 cm. Two waterlogging depths were established: 10 and 20 cm, and each treatment included four replicate plots. At the beginning of the experiment, the seedlings had three leaves, and the plant height was 5–7 cm. The growth of seedlings in the plot was regularly assessed.

### *Mowing and tilling to control* Spartina alterniflora

Five replicate plots were established in 2016 to carry out an experiment involving mowing together with tilling (hereafter referred to as MT) to control *Spartina alterniflora*. Each plot had an area of approximately 10 m$^2$, and the distance between the plots was more than 10 m. A control plot without MT was established near each MT plot. One fixed subplot with an area of one m$^2$ in each plot was set up for a later survey. At the end of the growing season (mid-October 2016), the aboveground plants of *Spartina alterniflora* were mowed and removed, and then the soil was tilled with a shovel. The tillage depth was approximately 20 cm, and the roots of *Spartina alterniflora* remained in the soil.

### *Spraying herbicides to control* Spartina alterniflora

In early June 2018, haloxyfop-r-methyl and glyphosate were sprayed onto the *Spartina alterniflora* canopy, the height of which was approximately 50 cm. Different doses of the herbicides were sprayed, each over an area of 100 m$^2$. The doses of haloxyfop-r-methyl were 0.15, 0.3 and 0.45 kg ha$^{-1}$ (hereafter referred to as H1, H2 and H3). The doses of glyphosate were 4.0 and 8.0 kg ha$^{-1}$ (hereafter referred to as G1 and G2). Three square plots with an area of one m$^2$ were randomly established for each vegetation survey. In the herbicide treatments, when the vast majority of *Spartina alterniflora* died, the survey area was expanded to four m$^2$.

## Field sampling and survey

The seasonal variation in the plant density and canopy height of *Spartina alterniflora* in the different plots were regularly investigated. Spike parameters, such as density and length, were investigated at the end of the growing season.

## Statistical analysis

One-way ANOVA was used to identify significant differences in the parameters of *Spartina alterniflora* among the various treatments. The parameters included plant density, canopy height, panicle density and so on. After testing for the homogeneity of variance

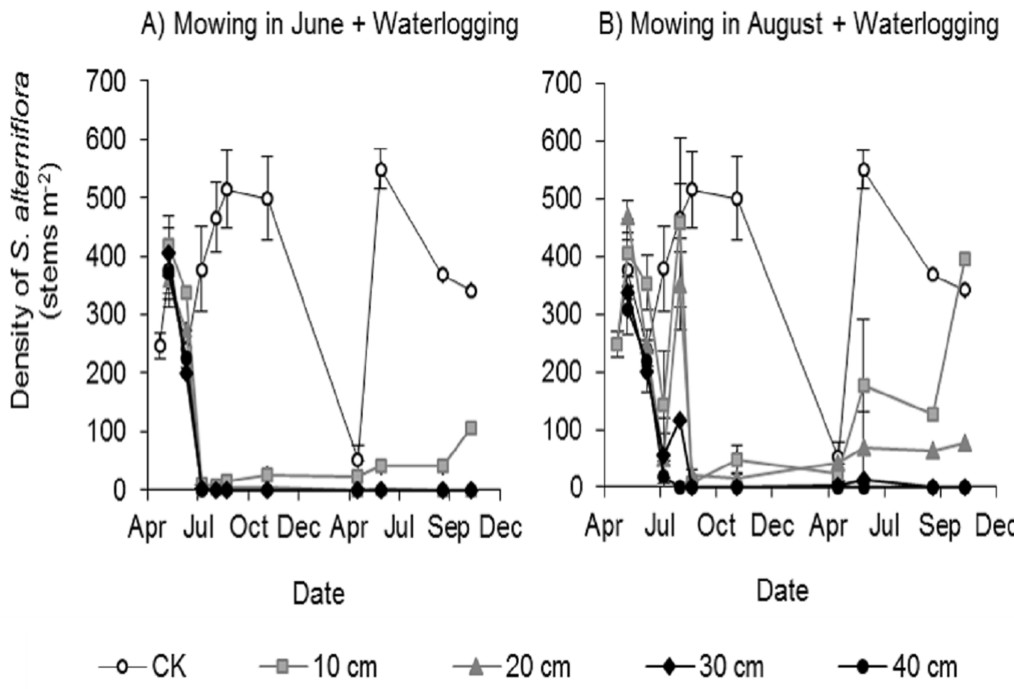

**Figure 3 Density of *Spartina alterniflora* under different combinations of mowing and waterlogging.** (A) Mowing and waterlogging in early June. (B) Mowing and waterlogging in early August. 10, 20, 30 and 40 cm indicate the various waterlogging levels. CK indicates the treatment with neither mowing nor waterlogging.

(Levene's test), the least significant difference method was used to carry out multiple comparison analysis. Significance for all statistical analyses was accepted at $p = 0.05$ level.

# RESULTS

## Control of *Spartina alterniflora* by mowing and waterlogging
### *Control of the asexual propagation of* Spartina alterniflora *by mowing and waterlogging*
*Effects of mowing and waterlogging on the density of* Spartina alterniflora

All mowing and waterlogging combinations significantly inhibited the germination of *Spartina alterniflora* (Fig. 3). The waterlogging at 30 and 40 cm depth after mowing in June or August completely inhibited the germination of *Spartina alterniflora*, with no *Spartina alterniflora* germinating from August 2017 to November 2018. The density of *Spartina alterniflora* in the 10 and 20 cm waterlogging treatments was less than 3.9% that in the CK treatment in 2017. The different mowing times also had an important effect on the germination of *Spartina alterniflora*. Under the same waterlogging level, Mow_6 resulted in better control than Mow_8. In the Mow_6 treatment, only one new ramet of *Spartina alterniflora* was found in the 12 plots at a 10 or 20 cm waterlogging depth. In the Mow_8 treatment, new ramets were found in two plots with a 10 cm waterlogging depth and three plots with a 20 cm waterlogging depth.

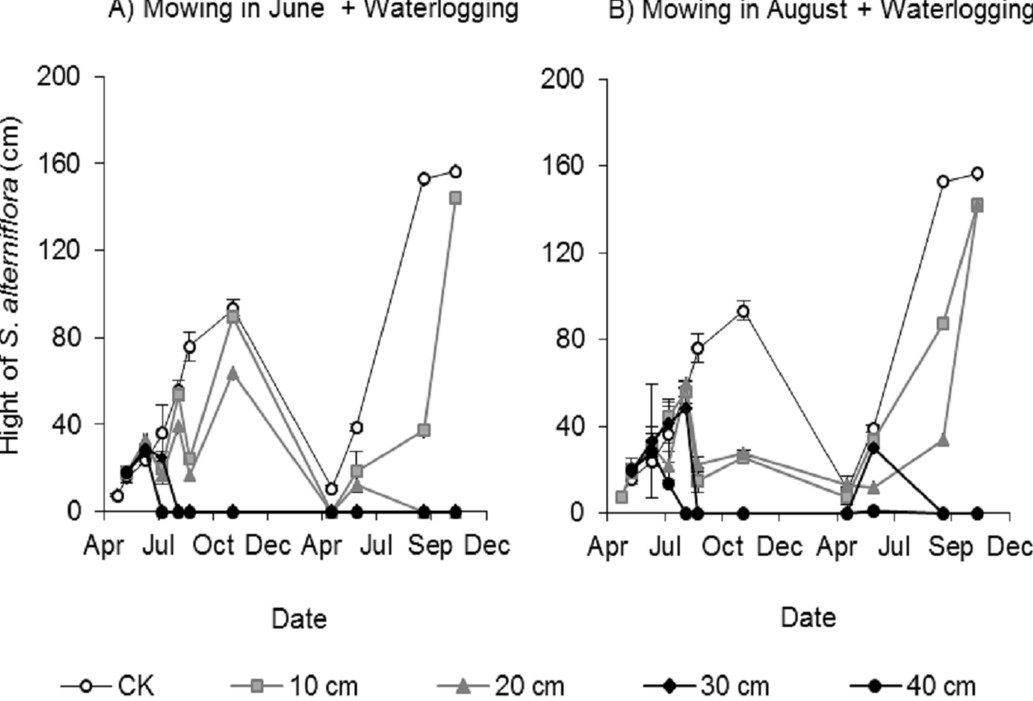

**Figure 4 Canopy height of _Spartina alterniflora_ under different combinations of mowing and waterlogging.** (A) Mowing and waterlogging in early June. (B) Mowing and waterlogging in early August. 10, 20, 30 and 40 cm indicate the various waterlogging levels. CK indicates the treatment without any mowing or waterlogging.

_Effects of mowing and waterlogging on the canopy height of_ Spartina alterniflora
Clonal ramets sprouted from the rhizomes of _Spartina alterniflora_ almost throughout the entire growing season; thus, we investigated the canopy height of _Spartina alterniflora_. The canopy height increased almost linearly in the control treatment and reached a maximum (93.1 ± 4.8 cm, mean ± standard error) in early November 2017. The height of the clonal ramets that regenerated after mowing was significantly affected by the waterlogging level. The regenerated clonal ramets in Mow_6 were taller than those in Mow_8 because of the longer growth time. At the end of the growing season in 2017, the height of the regenerated clonal ramets in the 10 and 20 cm waterlogging treatments in Mow_6 was 96% ($p > 0.1$) and 68% ($p < 0.01$), respectively, of the height in the CK treatment (Fig. 4A). The height of the regenerated clonal ramets in the 10 and 20 cm waterlogging treatments in Mow_8 was 27% ($p < 0.01$) and 30% ($p < 0.01$), respectively, of the height in CK (Fig. 4B). In 2018, the height in the 10 and 20 cm waterlogging treatments was close to that in CK at the end of the growing season. In addition, the height of _Spartina alterniflora_ in 2018 was much higher than that in 2017 ($p < 0.01$).

_Effects of different stubble heights on asexual reproduction control efficacy_
_Spartina alterniflora_ could be completely controlled if the stubble was waterlogged at a suitable water level after mowing. A single mowing could not effectively control _Spartina alterniflora_. The density of newly cloned ramets was approximately half that before mowing (Fig. 5A). Under the condition of long-term waterlogging at a level of 30 or 40 cm after

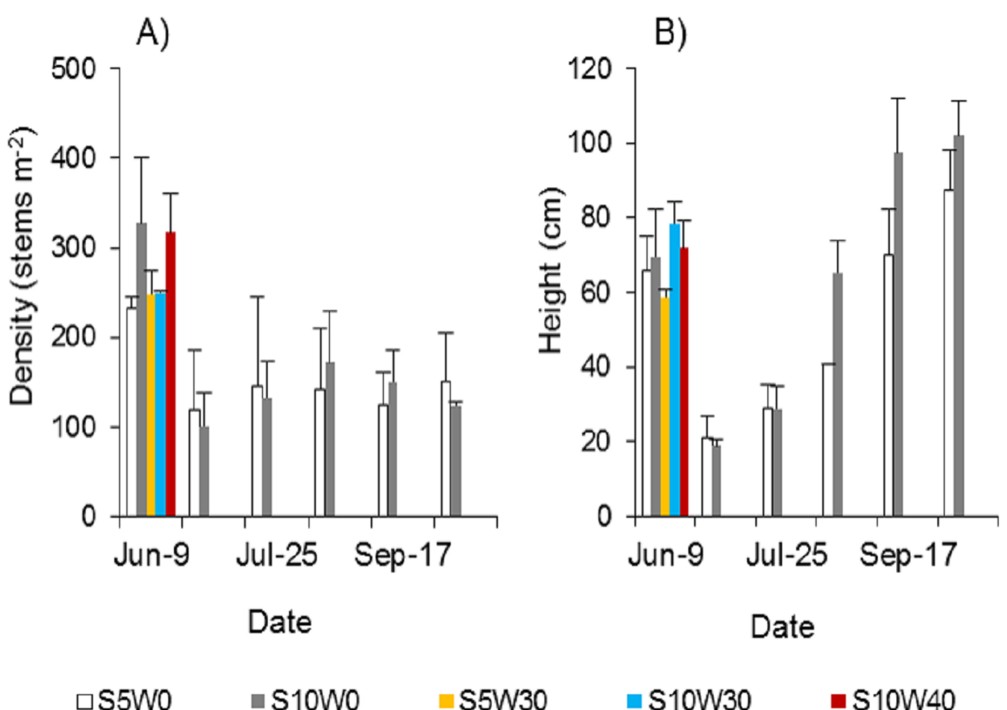

**Figure 5 Density (A) and canopy height (B) of *Spartina alterniflora* under different stubble height and waterlogging level treatments.** S5 W0 indicates that the height of the stubble (S) is five cm and the level of waterlogging (W) is zero cm, and the others are similar.

mowing, no ramets germinated regardless of whether the stubble height was five cm or 10 cm (Fig. 5A). The height of the stubble affected the growth of the new clonal ramets. Within 46 days after germination, the heights of the ramets showed no significant difference between the 10 and five cm stubble height treatments. However, 77 days after germination, the height of the ramets in the former treatment was 61% higher than that in the latter treatment ($p < 0.01$, Fig. 5B), and this difference was maintained until the end of the growing season.

### Control of the sexual propagation of Spartina alterniflora by mowing and waterlogging

*Effects of mowing and waterlogging on heading of* Spartina alterniflora

Because there were few regenerated clonal ramets after mowing and waterlogging, the spike density of *Spartina alterniflora* was far lower following those treatments than that in the control treatment at the end of the growing season in 2017. Among the 24 waterlogging plots under the Mow_6 treatment, spikes were found only in one 10 cm waterlogging plot and one 20 cm waterlogging plot. In the Mow_8 treatment, spikes were found only in one 10 cm waterlogging plot. The average spike density in the 10 cm and 20 cm plots was no more than 3.1% of that in the control treatment (Fig. 6A). The spike density of *Spartina alterniflora* in 2018 was similar to that in 2017 in all cases except the 10 and 20 cm waterlogging treatments in Mow_8 (Fig. 6D). The spike data indicate that the integrated approach could significantly inhibit the sexual propagation of *Spartina alterniflora* via seeds for 2 years.

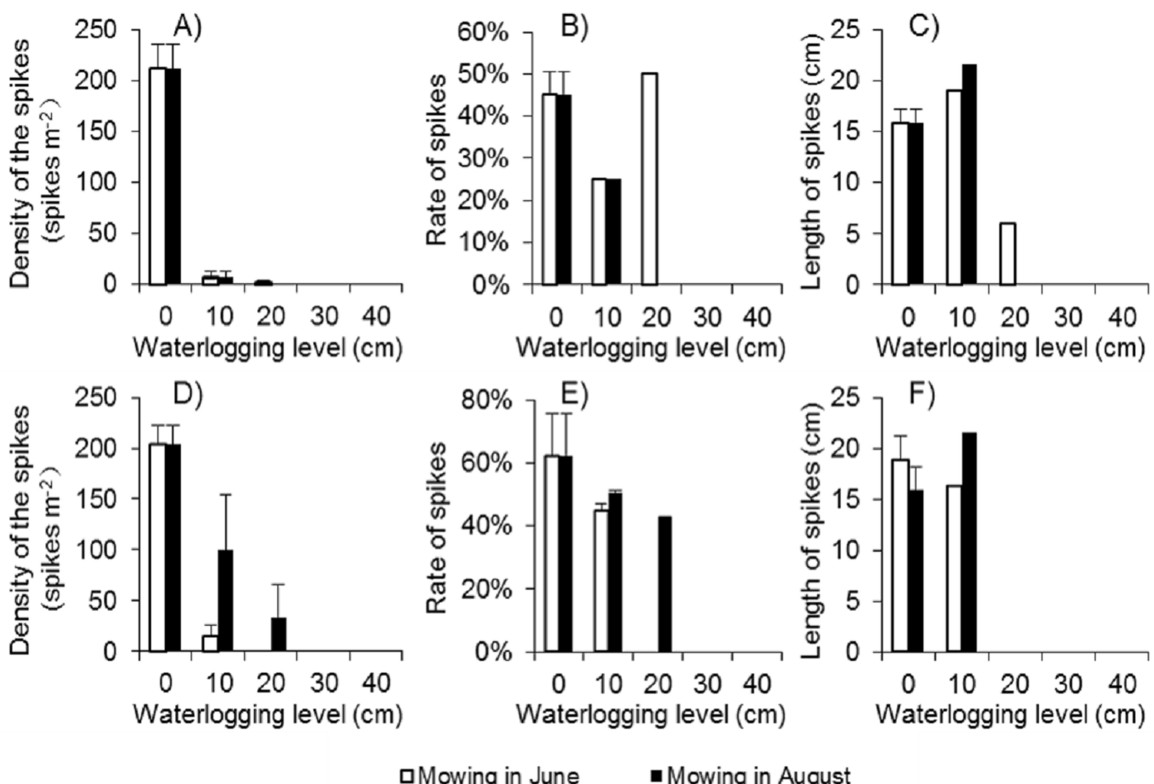

☐ Mowing in June ■ Mowing in August

**Figure 6 Spike parameters of *Spartina alterniflora* under different combinations of mowing and waterlogging in 2017 (A–C) and 2018 (D–F).** (A) Density of the spikes of *Spartina alterniflora* in 2017. (B) Rate of the spikes of *Spartina alterniflora* in 2017. (C) Length of the spikes of *Spartina alterniflora* in 2017. (D) Density of the spikes of *Spartina alterniflora* in 2018. (E) Rate of the spikes of *Spartina alterniflora* in 2018. (F) Length of the spikes of *Spartina alterniflora* in 2018.

The spike rate refers to the ratio of the number of plants with spikes to the number of all plants. The spike rate of *Spartina alterniflora* was 45% ± 6% in the control treatment in 2017. Waterlogging and mowing significantly decreased the spike rate (Fig. 6B). The spike rate in the 20 cm waterlogging plots in Mow_6 was similar to that in the control treatment. However, this value was not representative because there were only two plants in one of the 12 plots. In 2018, the spike rate was slightly higher than that in 2017. Moreover, the difference between the mowing and waterlogging treatments and the control treatment was smaller than that in the last year (Fig. 6E).

The length of the spike was not affected by 10 cm of waterlogging. A waterlogging level of 20 cm seemed to inhibit strike length; however, there was only one spike in this treatment, and its value might not be representative of all spikes (Figs. 6C and 6F).

*Effects of waterlogging on the growth of* Spartina alterniflora *seedlings*
At the beginning of the waterlogging experiment in late May, the seedling densities in the two treatments were very similar, at 159 ± 35 and 152 ± 34 stems m$^{-2}$ in the 10 cm and 20 cm waterlogging treatments, respectively (Fig. 7). After 15 days, many leaves turned yellow, and a few seedlings died, and the densities were 139 ± 28 and 146 ± 31 stems m$^{-2}$, respectively. All of the seedlings in the eight plots died 43 days after waterlogging.

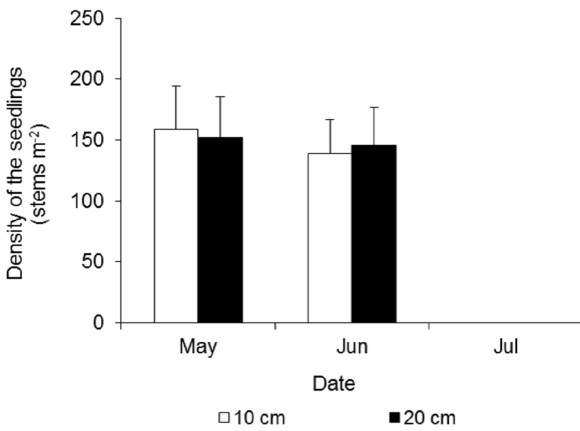

**Figure 7 Density dynamics of the seedlings of *Spartina alterniflora* under different waterlogging treatments.** Ten cm and 20 cm indicate the water level of waterlogging.

## Control of *Spartina alterniflora* by mowing and tilling

### *Effects of mowing and tilling on* Spartina alterniflora *density*

The sprouts of *Spartina alterniflora* include seedlings germinating from seeds and clonal ramets arising from rhizomes. In the early growing season, the seedlings from seeds were slender and grew very slowly, while the cloned ramets were robust and grew fast; thus, it was easy to distinguish them morphologically before July.

Mowing and tillage at the end of the growing season in 2016 almost completely inhibited the asexual reproduction of *Spartina alterniflora* in 2017 ($p < 0.001$). The density of cloned ramets in the MT treatment in early May 2017 was 2.4 plants m$^{-2}$, which was only 0.6% of that in the control treatment. One month later, the density of cloned ramets remained almost unchanged (2.6 plants m$^{-2}$, Fig. 8A). Although all the seeds of *Spartina alterniflora* were removed, the seeds of nearby *Spartina alterniflora* could enter the open MT plots via tidal or wind transportation. Thus, there were still some seedlings germinating from seeds in the MT plots. From May to June 2017, the seedling density in the MT treatment was lower than that in the CK treatment by 28–31% (Fig. 8B, $p > 0.05$). After July, it was impossible to distinguish seedlings from cloned ramets in terms of morphology. During the reproductive growing period (July to November 2017), the density of *Spartina alterniflora* in the MT treatment was 5–31% of that in the CK treatment ($p < 0.001$, Fig. 8C). *Spartina alterniflora* was completely restored in 2018. There was no significant difference in the density of *Spartina alterniflora* between the MT and CK treatments during the reproductive growing period in 2018 (Figs. 8D–8F).

### *Effects of mowing and tilling on the canopy height of* Spartina alterniflora

Mowing and tillage effectively inhibited both the germination and growth of *Spartina alterniflora* the following year (Fig. 9). The clonal plant height in the MT and CK treatments increased continuously, but the growth rate of clonal ramets was significantly inhibited by MT, and the canopy height in the MT treatment was always 25–45% of that in the CK treatment ($p < 0.001$, Fig. 9A). From germination to early June in 2017, the

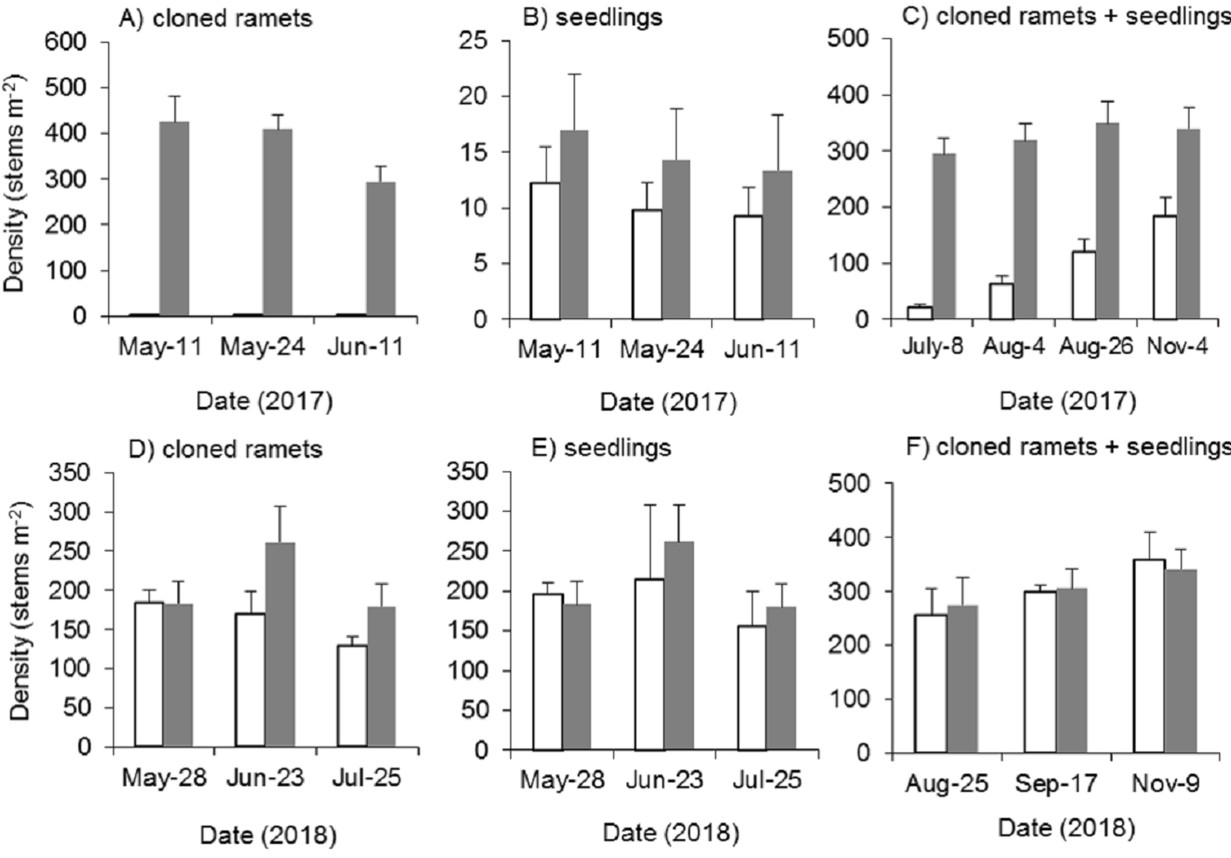

**Figure 8 Dynamics of the density of *Spartina alterniflora* after mowing and tilling.** (A) Density of cloned ramets in 2017. (B) Density of seedlings in 2017. (C) Density of cloned ramets and seedlings in 2017. (D) Density of cloned ramets in 2018. (E) Density of seedlings in 2018. (F) Density of cloned ramets and seedlings in 2018. The white bars indicate mowing and tilling. The grey bars indicate the control treatment with neither mowing nor tillage.

seedlings grew very slowly, plant height remained below seven cm and there was no significant difference in seedling height between the MT and CK treatments (Fig. 9B). During the reproductive growth period, the canopy height of *Spartina alterniflora* in the MT treatment remained much lower than that in the CK treatment. At the end of the growing season in early November, the canopy height in the CK treatment was 105.7 cm, which was higher than that in the MT treatment by 53.4% ($p < 0.001$). Therefore, mowing and tilling inhibited both the germination and growth of *Spartina alterniflora* in the following year. However, the growth rate of *Spartina alterniflora* was well restored in 2018, and the canopy height in the MT treatment was very close to that in the CK treatment, especially during the reproductive period (Figs. 9D–9F).

### *Effects of mowing and tilling on the heading of* Spartina alterniflora
Mowing and tilling at the end of the growing season significantly inhibited the spike density, spike rate and spike length of *Spartina alterniflora* in the following year (Fig. 10). The spike density of *Spartina alterniflora* in the MT treatment was only 21.9% ($p < 0.01$) of that in the CK treatment. Tillage also inhibited the growth of *Spartina alterniflora*,

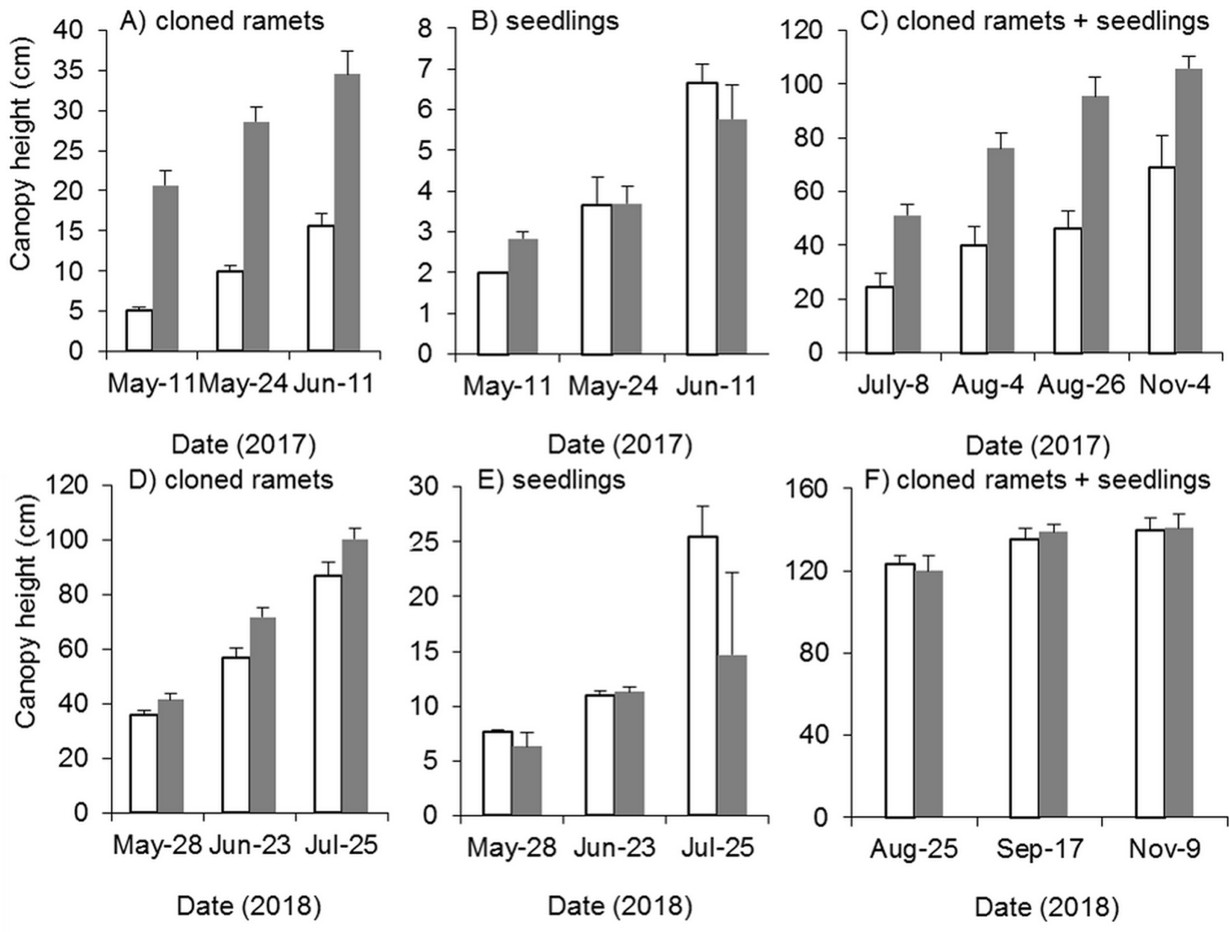

**Figure 9 Dynamics of the canopy height of *Spartina alterniflora* after mowing and tilling.** (A) Canopy height of cloned ramets in 2017. (B) Canopy height of seedlings in 2017. (C) Canopy height of cloned ramets and seedlings in 2017. (D) Canopy height of cloned ramets in 2018. (E) Canopy height of seedlings in 2018. (F) Canopy height of cloned ramets and seedlings in 2018. The white bars indicate mowing and tilling. The grey bars indicate the control treatment with neither mowing nor tillage.

resulting in a lower spike rate, which was the ratio of the number of spikes to the number of *Spartina alterniflora* stems. The growth of spikes was also inhibited, and the spike length in the MT treatment was shorter than that in the CK treatment by 13% ($p < 0.05$). The spike data indicated that the seed yield in the following year was highly inhibited by mowing and tilling. As a result, the sexual propagation of *Spartina alterniflora* will be continuously inhibited.

Due to the restoration of asexual reproductive capacity and the secondary invasion of seeds, *Spartina alterniflora* in the MT treatment grew very well in 2018, and its panicle growth features were even better than those in the CK treatment (Fig. 10).

## Control of *Spartina alterniflora* by herbicides
### Effects of herbicides on the growth of *Spartina alterniflora*
The 0.15–0.45 kg ha$^{-1}$ dose of haloxyfop-r-methyl had a strong weed control effect. Haloxyfop-r-methyl was sprayed with a backpack sprayer in early June 2018. One and a

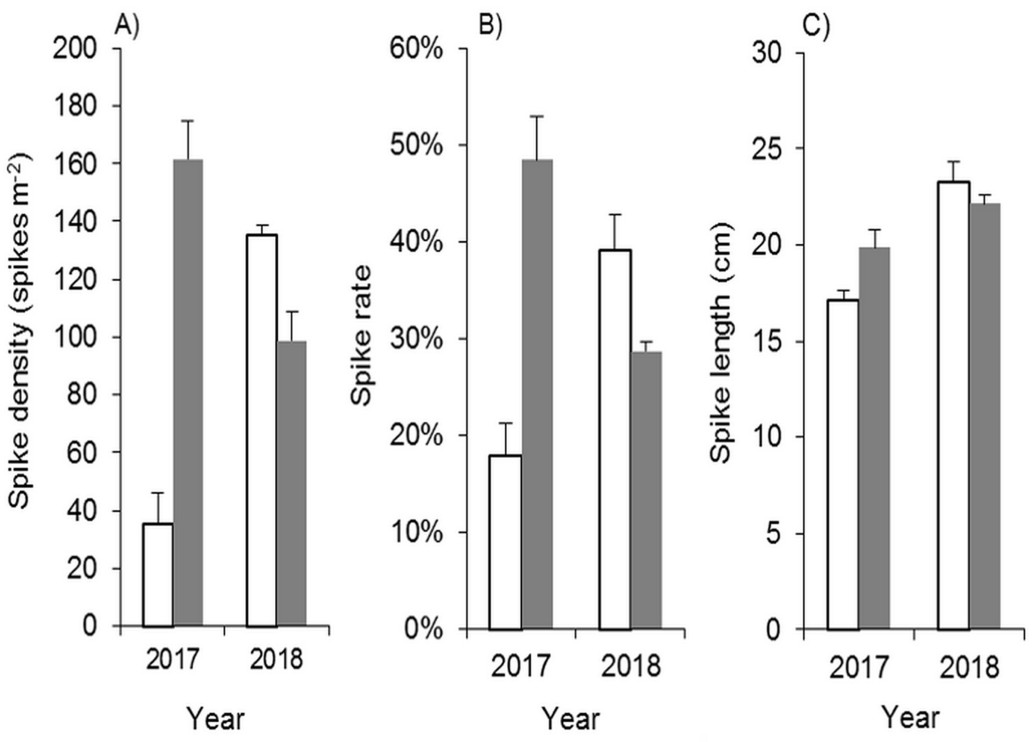

**Figure 10 Effects of mowing and tilling on the heading of *Spartina alterniflora*.** The white bars indicate mowing and tilling. The grey bars indicate the control treatment with neither mowing nor tillage. (A) Spike density. (B) Spike rate. (C) Spike length.

half months later, the vast majority of *Spartina alterniflora* were dead (Fig. 11A). Although all three doses of haloxyfop-r-methyl had good herbicidal effects, there were significant differences among the different doses. At the end of the growing season, in comparison to the control treatment, the density of *Spartina alterniflora* in the H1, H2 and H3 treatments decreased by 58.6% ($p < 0.01$), 98.3% ($p < 0.01$) and 99.5% ($p < 0.01$), respectively. The application of 0.3 and 0.45 kg ha$^{-1}$ haloxyfop-r-methyl achieved a perfect weed control effect. Although a small number of *Spartina alterniflora* survived in haloxyfop-r-methyl plots, their growth was significantly inhibited. The plant height of *Spartina alterniflora* in haloxyfop-r-methyl plots was approximately half that in the CK treatment at the end of the growing season ($p < 0.05$, Fig. 11B).

The control effect of glyphosate on *Spartina alterniflora* was inferior to that of haloxyfop-r-methyl. At the end of the growing season, the density of *Spartina alterniflora* in the G1 and G2 treatments decreased by 16.1% ($p > 0.1$) and 23.4% ($p > 0.05$), respectively, and the canopy height in the G1 and G2 treatments was shorter than that in the CK treatment by 36.8% ($p < 0.01$) and 44.6% ($p < 0.01$), respectively.

### Effects of herbicides on the heading of Spartina alterniflora

In the herbicide treatments, the surviving *Spartina alterniflora* grew slowly, and their spikes were also poisoned by herbicides. The spike density of *Spartina alterniflora* in the haloxyfop-r-methyl treatments was only 0.6–16.4% of that in the CK treatment

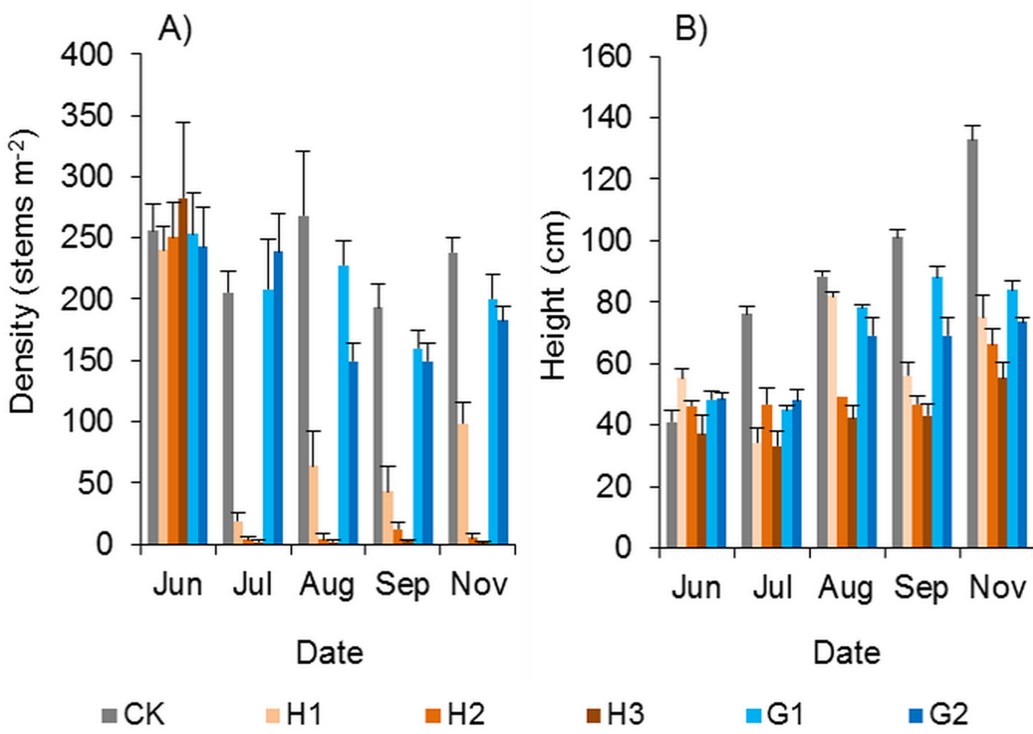

**Figure 11 Effects of herbicides on the growth of *Spartina alterniflora*.** (A) Density of *Spartina alterniflora*. (B) Canopy height of *Spartina alterniflora*. CK indicates the control treatment without herbicide. H1, H2 and H3 indicate the treatments with 0.15, 0.30 and 0.45 kg ha$^{-1}$ haloxyfop-r-methyl, respectively. G1 and G2 indicate the treatments with 4.0 and 8.0 kg ha$^{-1}$ glyphosate, respectively.

($p < 0.01$, Fig. 12A). However, there were no significant differences among the different doses. The spike length of *Spartina alterniflora* in the H1, H2 and H3 treatments was 88.3% ($p < 0.1$), 65.4% ($p < 0.01$) and 36.1% ($p < 0.01$) of that in the CK treatment, respectively (Fig. 12B). However, there were no significant differences among the different doses. Panicle development was severely inhibited in the H2 and H3 treatments, and there were no mature seeds. Therefore, the sexual reproduction of *Spartina alterniflora* was completely inhibited by the two treatments.

Glyphosate significantly inhibited the spike density of *Spartina alterniflora*, and the spike density in the G1 and G2 treatments was 41.0% and 12.7% of that in the CK treatment ($p < 0.01$, Fig. 12A). However, the growth of spikes in the G1 and G2 treatments was good. The spike length was almost the same as that in the CK treatments, and many mature seeds were produced, which indicated that glyphosate application could not satisfactorily inhibit the sexual reproduction of *Spartina alterniflora* in the following year.

## Cost of controlling *Spartina alterniflora*

On the basis of our experimental study, 200 m$^2$ of *Spartina alterniflora* were treated with each control approach to preliminarily estimate the control cost. The control costs of mowing + 30 cm waterlogging, mowing + tillage and spraying 0.3 kg ha$^{-1}$ of haloxyfop-r-methyl were 4,104, 3,284 and 1,067 dollars per hectare, respectively. Because these
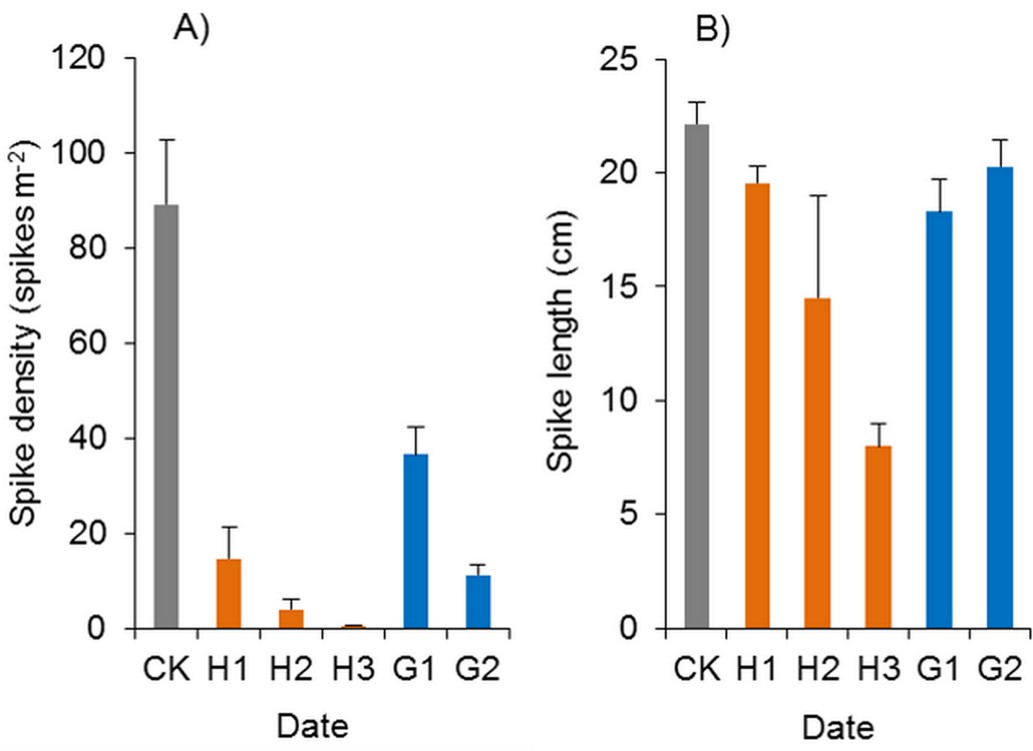

**Figure 12 Effects of haloxyfop-r-methyl on the spikes of *Spartina alterniflora*.** (A) Spike density of *Spartina alterniflora*. (B) Spike length of *Spartina alterniflora*. CK indicates the control treatment without herbicide. H1, H2 and H3 indicate the treatments with 0.15, 0.30 and 0.45 kg ha$^{-1}$ haloxyfop-r-methyl, respectively. G1 and G2 indicate the treatments with 4.0 and 8.0 kg ha$^{-1}$ glyphosate, respectively.

approaches were nearly 100% effective in eliminating *Spartina alterniflora*, we assumed that 10% of the cost would be used for maintenance or supplementary control in the second year.

## DISCUSSION

### Control efficacy of mowing and waterlogging

Mowing can prevent the photosynthesis of *Spartina alterniflora*, and waterlogging after mowing may lead to the gradual death of roots due to hypoxia (*Xie et al., 2018*). Many studies have shown that mowing in addition to waterlogging can eradicate *Spartina alterniflora* (Table 1). This study aimed to improve control effectiveness of this integrated approach and reduce its cost.

The control efficacy of mowing is closely related to mowing timing. Improper timing of mowing, especially during the later growing season, may promote the regeneration of *Spartina alterniflora* (*Tan et al., 2010*). The sprouts of *Spartina alterniflora* come from seed germination and rhizome cloning. The clonal reproduction of rhizomes occurs almost throughout the entire growing season. From the germination stage, the density of *Spartina alterniflora* increased gradually. Due to the death of the seedlings germinating from seeds, the plant density began to decline in May and reached a minimum in early June.

**Table 1 A summary of the control efficacy of mowing plus waterlogging.**

| No. | Details | Survey time | Control efficacy[#] (%) | Experimental site | References |
|-----|---------|-------------|------------------|-------------------|------------|
| 1 | Mowing + 10 cm waterlogging in early June 2017 | Nov-17<br>Nov-18 | 95<br>69 | YRE[*], Shandong, China | This study |
| 2 | Mowing + 10 cm waterlogging in early August 2017 | Nov-17<br>Nov-18 | 90<br>−16 | YRE, Shandong, China | This study |
| 3 | Mowing + 20 cm waterlogging in early June 2017 | Nov-17<br>Nov-18 | 100<br>100 | YRE[*], Shandong, China | This study |
| 4 | Mowing + 20 cm waterlogging in early August 2017 | Nov-17<br>Nov-18 | 99<br>100 | YRE, Shandong, China | This study |
| 5 | Mowing + 30 or 40 cm waterlogging in early June or August 2017 | Nov-17<br>Nov-18 | 100<br>100 | YRE, Shandong, China | This study |
| 6 | Mowing from late June to early July + 60–70 cm waterlogging in 2011 | Oct-11<br>Oct-12 | 100<br>100 | Yangtze River Estuary, Shanghai, China | (Sheng et al., 2014) |
| 7 | Mowing + 50–60 cm waterlogging in 2008 | Oct-12 | 100 | Yangtze River Estuary, Shanghai, China | (Sheng et al., 2014) |
| 8 | Mowing + 30–50 cm waterlogging in March + mowing in July 2007 | Oct-07<br>Oct-08 | 100<br>100 | Yangtze River Estuary, Shanghai, China | (Yuan et al., 2011) |

Notes:
[#] Density reduction ratio of *Spartina alterniflora* compared with the control treatment.
[*] YRE indicates the Yellow River Estuary.

After that, there were no newly germinated seeds, and the density of *Spartina alterniflora* increased continuously due to the enhancement of the clonal reproductive ability of rhizomes (CK in Fig. 3). The seasonal variation in plant density indicates that the clonal reproductive capacity of the *Spartina alterniflora* community may be weakest in early June. Therefore, early June, that is, the end of the vegetative growth period, may be the best time to control the asexual reproduction of *Spartina alterniflora*. Our study confirmed that mowing in early June is more effective than mowing in early August under the same waterlogging conditions (Fig. 3). However, differences in climate or topography may lead to differences in optimal mowing times. In the Yangtze Estuary, *Tang et al. (2009)* found that mowing during the flowering stage in early July had a better control effect than mowing during other periods. *Yuan et al. (2011)* also found that mowing along with waterlogging during the flowering stage in early July can eradicate *Spartina alterniflora* in the Yangtze Estuary. In summary, the optimal time for mowing is from the end of vegetative growth to the flowering stage.

Although *Spartina alterniflora* has strong resistance to flooding, continuous waterlogging stress will inhibit its growth. The control effect of waterlogging is closely related to the phenological phase and water level. The effect of continuous waterlogging on seedlings is greater than that on ramets. Our results showed that 10–20 cm waterlogging killed all of the seedlings, which were 5–7 cm high in 43 days. *Chen et al. (2011)* also found that continuous waterlogging at a depth of 20 cm could lead to the death of seedlings (height 7–10 cm) within 3 months. This is likely due to the poor resistance of seedlings to

waterlogging. We observed that the seedlings of *Spartina alterniflora* were very slim and grew slowly. Two months after germination, the height of seedlings was still less than seven cm, and the number of leaves was no more than three.

The water level of waterlogging has a great influence on the control efficacy. The ramets of *Spartina alterniflora* grow fast and have strong adaptability to waterlogging stress. When the level of waterlogging is lower than the height of plants, the chlorophyll content in the leaves increases significantly, and the growth of *Spartina alterniflora* is promoted. When water levels are higher than the plants, waterlogging leads to a significant decline in chlorophyll content and photosynthetic rate and eventually significantly inhibits the growth and reproduction of *Spartina alterniflora* (*Mateos-Naranjo et al., 2007*; *Yuan & Zhang, 2010*). Even if the water level is as high as 100 cm, it is very difficult to completely kill *Spartina alterniflora* by waterlogging alone (*Yuan & Zhang, 2010*). The integrated method of mowing and waterlogging can achieve better control efficacy. Studies in the Yangtze Estuary of China have found that *Spartina alterniflora* can be eradicated by waterlogging at 30–70 cm after mowing (*Sheng et al., 2014*; *Yuan et al., 2011*). Our study in the Yellow River Estuary of China found that the waterlogging level after mowing could be reduced to 20 cm if *Spartina alterniflora* was controlled at the end of the vegetative growth period. The control efficacy of *Spartina alterniflora* could be as high as 90% even if the waterlogging level was only 10 cm after mowing. Our previous studies showed that the rhizomes of *Spartina alterniflora* were dead after mowing and waterlogging at a level of 20 cm for 4.5 months (*Xie et al., 2018*).

In summary, the integration of mowing and waterlogging can eradicate *Spartina alterniflora*, and the timing of mowing from the late vegetative growth stage to the flowering stage of *Spartina alterniflora* is suitable, but the depth of waterlogging may vary in different regions. A lower waterlogging level implies easier control and lower cost. Therefore, it is better to conduct experimental research before the large-scale control of *Spartina alterniflora* is implemented.

## Control efficacy of mowing and tilling

There are few studies on controlling *Spartina alterniflora* by mowing and tilling (Table 2). Mowing is the pretreatment of tilling, and tilling plays the main role in control. *Spartina alterniflora* enters dormancy in the cold winter. Tillage can destroy the rhizome and make it vulnerable to cold and the tide and therefore easily affects the asexual reproductive capacity of the rhizome. Therefore, the asexual reproduction of *Spartina alterniflora* was almost completely inhibited the following spring (Fig. 8A). However, if the secondary invasion of seeds is unavoidable, the control area will be reoccupied by *Spartina alterniflora* after a certain period of time (Figs. 8D–8F).

The choice of tillage time may have a great influence on control efficacy. This study suggested that mowing and tilling at the end of the growing season reached a satisfactory control efficacy. As described in the section "Control efficacy of mowing and waterlogging", the asexual reproductive capacity of *Spartina alterniflora* may be weakest at the end of the vegetative growth stage, so tillage during this period may also achieve very good control efficacy. In the coastal zone of Fujian Province of China, mowing and tillage

**Table 2 A summary of the control efficacy of mowing plus tilling.**

| No. | Details | Survey time | Control efficacy[#] (%) | Experimental site | Reference |
|-----|---------|-------------|-------------------------|-------------------|-----------|
| 1 | Mowing and tilling in late October 2016 | Jun-17 Nov-17 Nov-18 | 99 46 −5 | YRE*, Shandong, China | This study |
| 2 | Mowing in early July 2006, tilling when the new seedlings grew to 10–15 cm | Oct-06 Oct-07 | 100 100 | Quanzhou Bay, Fujian, China | (*Tan, 2008*) |
| 3 | Winter tilling | – | 73 | Willapa Bay, Louisiana, USA | (*Patten, 2004*) |

Notes:
[#] Density reduction ratio of *Spartina alterniflora* compared with the control treatment.
\* YRE indicates Yellow River Estuary.
– No available information.

in early July could eradicate *Spartina alterniflora*, which did not reappear in the second and third years (*Tan, 2008*). Some studies have reported that the control efficiency of winter tillage is only 73% (*Patten, 2004*). If the rhizome of *Spartina alterniflora* is further broken after tillage, the control efficiency may be improved (*Mateos-Naranjo et al., 2012*).

## Control efficacy of herbicides

Chemical control is usually carried out by applying herbicides to eradicate *Spartina alterniflora*. Many herbicides, such as haloxyfop-r-methyl, glyphosate, glufosinate ammonium and imazapyr, have been used to control *Spartina alterniflora*. The control efficacy of different herbicides varies greatly, ranging from 12% to 100% (Table 3). The U. S. Environmental Protection Agency only allows glyphosate and imazapyr to be used in estuarine environments (*Knott, Webster & Nabukalu, 2013*). Glyphosate is a widely used herbicide, but its efficacy for control of *Spartina alterniflora* is unsatisfactory, the vast majority of which is less than 62% (*Knott, Webster & Nabukalu, 2013*; *Mateos-Naranjo et al., 2012*; *Patten, 2004*). Although some studies have shown that the control efficacy of haloxyfop-r-methyl against *Spartina alterniflora* is very poor, the present study found that the control efficacy of haloxyfop-r-methyl is close to 100% at an appropriate dose. In addition to the dosage of herbicides, the growth period of *Spartina alterniflora* also affects the control efficacy of herbicides. For example, the same dosage of glyphosate showed 93% control efficacy against *Spartina alterniflora* seedlings but only 16–25% control efficacy against mature plants (*Knott, Webster & Nabukalu, 2013*). Glufosinate and imazapyr have better mortality effects on the seedlings of *Spartina alterniflora*, but their control effect against mature *Spartina alterniflora* is also poor, usually less than 33% (*Knott, Webster & Nabukalu, 2013*; *Patten, 2004*). The control efficacy of herbicides is also influenced by wind, tidal cycles and sediment cover on stems and leaves (*Hedge, Kriwoken & Patten, 2003*).

It must be noted that it is almost impossible to achieve 100% control efficacy by spraying herbicides due to the mutual occlusion of dense stems and leaves or uneven spraying. In this study, 0.30 kg ha$^{-1}$ haloxyfop-r-methyl could kill 98% of *Spartina alterniflora*. when increasing the dosage of from 0.30 to 0.45 kg ha$^{-1}$, there were still a few surviving

**Table 3 A summary of the control efficacy of spraying herbicides.**

| No. | Herbicide | Details | Survey time | Control efficacy[#] (%) | Experimental site | References |
|---|---|---|---|---|---|---|
| 1 | Haloxyfop-r-methyl | 0.15 kg ha$^{-1}$ in early June 2018 | Nov-18 | 59 | YRE, Shandong, China | This study |
| 2 | Haloxyfop-r-methyl | 0.30 kg ha$^{-1}$ in early June 2018 | Nov-18 | 98 | YRE, Shandong, China | This study |
| 3 | Haloxyfop-r-methyl | 0.45 kg ha$^{-1}$ in early June 2018 | Nov-18 | 99 | YRE, Shandong, China | This study |
| 4 | Haloxyfop-r-methyl | A total of 0.19 kg ha$^{-1}$, sprayed three times in June, July and September 2011 | Oct-12 | 12 | Yangtze River Estuary, Shanghai, China | (*Sheng et al., 2014*) |
| 5 | Glyphosate | 4.0 kg ha$^{-1}$ in early June 2018 | Nov-18 | 16 | YRE, Shandong, China | This study |
| 6 | Glyphosate | 8.0 kg ha$^{-1}$ in early June 2018 | Nov-18 | 23 | YRE, Shandong, China | This study |
| 7 | Glyphosate | 1.06 or 2.13 kg ha$^{-1}$, 13 cm seedlings in early May 2011 | Jul-11 | 95 | Louisiana, USA | (*Knott, Webster & Nabukalu, 2013*) |
| 8 | Glyphosate | 1.06 kg ha$^{-1}$, 60 cm plants in early May 2011 | Jul-11 | 16[*] | Louisiana, USA | (*Knott, Webster & Nabukalu, 2013*) |
| 9 | Glyphosate | 2.13 kg ha$^{-1}$, 60 cm plants in early May 2011 | Jul-11 | 25[*] | Louisiana, USA | (*Knott, Webster & Nabukalu, 2013*) |
| 10 | Glyphosate | 7.2 kg ha$^{-1}$, the end of growing season in December 2006 | 2007 | 62 | Atlantic coast of southwestern Spain | (*Mateos-Naranjo et al., 2012*) |
| 11 | Glyphosate | 9.0 kg ha$^{-1}$, in 2002 or 2003 | 2008 – | 48 31 | Willapa Bay, Louisiana, USA | (*Patten, 2004*) |
| 12 | Glyphosate | 38.0 kg ha$^{-1}$, in 2002 or 2003 | – | 57 | Louisiana, USA | (*Patten, 2004*) |
| 13 | Glufosinate | 0.82 or 1.64 kg ha$^{-1}$, 13 cm seedlings in early May 2011 | Jul-11 | 100 | Louisiana, USA | (*Knott, Webster & Nabukalu, 2013*) |
| 14 | Glufosinate | 0.82 kg ha$^{-1}$, 60 cm plants in early May 2011 | Jul-11 | 19.7[*] | Louisiana, USA | (*Knott, Webster & Nabukalu, 2013*) |
| 15 | Glufosinate | 1.64 kg ha$^{-1}$, 60 cm plants in early May 2011 | Jul-11 | 25.4[*] | Louisiana, USA | (*Knott, Webster & Nabukalu, 2013*) |
| 16 | Imazapyr | 1.05 or 2.11 kg ha$^{-1}$, 13 seedlings cm in early May 2011 | Jul-11 | 93 | Louisiana, USA | (*Knott, Webster & Nabukalu, 2013*) |
| 17 | Imazapyr | 1.05 kg ha$^{-1}$, 60 cm plants in early May 2011 | Jul-11 | 33[*] | Louisiana, USA | (*Knott, Webster & Nabukalu, 2013*) |
| 18 | Imazapyr | 2.11 kg ha$^{-1}$, 60 cm plants in early May 2011 | Jul-11 | 32[*] | Louisiana, USA | (*Knott, Webster & Nabukalu, 2013*) |
| 19 | Imazapyr | 1.68 kg ha$^{-1}$, 170 cm plants in August 2003 | – | 82 | Louisiana, USA | (*Patten, 2004*) |

**Notes:**
[#] Density reduction ratio of *Spartina alterniflora* compared with the control treatment.
[*] The proportion of damaged stems, no dead stems.
– No information.

plants in some areas (1–5 stems m$^{-2}$ on average). Other studies have found similar results. The application of glyphosate in the same location for several consecutive years did not improve the overall control significantly compared with the application of glyphosate for only one year, and there were still several *Spartina alterniflora* plants per square metre (*Patten, 2004*). This sporadic survival of *Spartina alterniflora* may lead to large-scale secondary invasion over the next few years. It is not enough to spray herbicides only once.

The surviving *Spartina alterniflora* should be sprayed with herbicides for a second or even a third time.

Chemical methods for controlling *Spartina alterniflora* are likely to have negative effects. On the one hand, chemical agents usually cause some residual toxicity; on the other hand, they often cause harm to other plants and animals, thereby destroying the local soil and water ecosystems (*Kilbride & Paveglio, 2001*; *Paveglio et al., 1996*). *Qiao et al. (2019)* found that the crab density was significantly lower than that in the control treatment after spraying haloxyfop-r-methyl or glyphosate for 4 months, but the crab population recovered 11 months after spraying herbicides. However, many studies have found that herbicides are not harmful organisms on beaches or in estuaries (*Patten, 2003*; *Shimeta et al., 2016*). Species richness and diversity value of native plants were not affected by glyphosate (*Mateos-Naranjo et al., 2012*). Our study found that haloxyfop-r-methyl had no effect on *Suaeda salsa*, a native vegetation. This harmlessness may be due to low-dose exposure to herbicides because herbicides are mainly taken up by plant stems and leaves or washed away by tidal water, and only a small amount reaches sediments (*Shimeta et al., 2016*). The influence of herbicides on the environment is closely related to the amount and time of herbicide application. In the future, the optimal time and minimum dosage of herbicides should be evaluated to minimize their negative effects on the environment.

## Cost of controlling *Spartina alterniflora*

The cost of controlling *Spartina alterniflora* is very high. The mean annual cost to manage Willapa *Spartina alterniflora* during five years was ~$3,500 km$^{-1}$ of shoreline per year (*Patten, O'Casey & Metzger, 2017*). The cost of chemical control in this study was 1,067 dollars per hectare. By comparison, the average annual cost for the chemical control of *Spartina alterniflora* was 2,414 dollars per hectare in South Africa (*Riddin, van Wyk & Adams, 2016*). The control costs of mowing and waterlogging in this study were 4,104 dollars per hectare. *Yuan et al. (2011)* reported that the control cost of mowing and waterlogging was only 500 dollars per hectare. Because of the higher waterlogging level than that in this study and the need for pumping, the low cost reported by Yuan is almost impossible.

## CONCLUSION

This study covers a variety of approaches to control *Spartina alterniflora*, and all of the approaches can achieve very high control effectiveness. This study also provides a possibility to reduce the cost of controlling *Spartina alterniflora*.

The integrated approach of mowing and waterlogging can completely inhibit the sexual and asexual reproduction of *Spartina alterniflora*, thus achieving the goal of eradicating *Spartina alterniflora*. It is recommended that this method be used in areas with frequent tidal flooding. The technical details of this approach include (1) mowing *Spartina alterniflora* at the end of the vegetative growth stage, with the height of the stubble being less than 10 cm, and (2) continuous waterlogging after mowing until the end of the year, with the water level being 20–30 cm.

The integrated approach of mowing and tilling at the end of the growing season can almost completely inhibit the asexual reproduction of *Spartina alterniflora*, and the removal of *Spartina alterniflora* can effectively inhibit its sexual reproduction by removing seeds from the system. Therefore, this integrated approach is a good way to control *Spartina alterniflora*. This method may be more suitable for places with cold winters. In addition, if we want to extend this approach, special machinery suitable for muddy tidal flats is needed.

The application of haloxyfop-r-methyl at a dose of 0.3–0.45 kg ha$^{-1}$ can almost eradicate *Spartina alterniflora*, and its control efficacy was more than 98%. The control efficacy of glyphosate at a dose of 4.0–8.0 kg ha$^{-1}$ was less than 23%. Therefore, haloxyfop-r-methyl can be used to control *Spartina alterniflora*. To minimize the potential environmental pollution, herbicides are recommended for new invasive patches of *Spartina alterniflora*.

Finally, it is difficult to eradicate *Spartina alterniflora* only once. Later investigation and re-control for the remaining *Spartina alterniflora* are probably necessary. Large-scale control is also needed. Otherwise, the controlled areas are likely to be re-invaded by the seeds of *Spartina alterniflora* from seawater in a few years.

### Funding
This research was funded by the National Nature Science Foundation of China (41671089), the Science and Technology Service Network Initiative (KFJ-STS-ZDTP-023), and the Special Funds of National Nature Reserve (Y891061021, Y639071021). There was no additional external funding received for this study. The funders had no role in study design, data collection and analysis, decision to publish, or preparation of the manuscript.

### Grant Disclosures
The following grant information was disclosed by the authors:
National Nature Science Foundation of China: 41671089.
Science and Technology Service Network Initiative: KFJ-STS-ZDTP-023.
Special Funds of National Nature Reserve: Y891061021 and Y639071021.

### Competing Interests
The authors declare that they have no competing interests.

### Author Contributions
- Baohua Xie conceived and designed the experiments, performed the experiments, analyzed the data, contributed reagents/materials/analysis tools, prepared figures and/or tables, authored or reviewed drafts of the paper, approved the final draft.
- Guangxuan Han conceived and designed the experiments, analyzed the data, contributed reagents/materials/analysis tools, authored or reviewed drafts of the paper, approved the final draft.

- Peiyang Qiao performed the experiments, prepared figures and/or tables, approved the final draft.
- Baoling Mei conceived and designed the experiments, approved the final draft.
- Qing Wang performed the experiments, contributed reagents/materials/analysis tools, approved the final draft.
- Yingfeng Zhou performed the experiments, approved the final draft.
- Anfeng Zhang performed the experiments, approved the final draft.
- Weimin Song performed the experiments, approved the final draft.
- Bo Guan performed the experiments, approved the final draft.

### Data Availability

The raw data are available in the Supplemental File.

### Supplemental Information

Supplemental information for this article can be found online at http://dx.doi.org/10.7717/peerj.7655#supplemental-information.

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
