# Peer review of "Effects of mechanical and chemical control on invasive Spartina alterniflora in the Yellow River Delta, China"

_PeerJ, doi:10.7717/peerj.7655_

## Round 0.1 · original submission · Minor Revisions

Dear authors,

This manuscript presents such interesting data on the potential mechanical and chemical control on Spartina alterniflora invasion in China. The three reviewers were quite positive to the way the manuscript was presented. Most of the reviewers’ minor concerns were related to the English editing, some literature updating in the introduction section and the presentation of the discussion. Therefore, whether the authors would be able to provide a new version of the manuscript that incorporates the reviewer's concerns they will have an improved version of it potentially being a well worth paper to be published in PeerJ.

Here are some specific points stressed below that should be seriously taken into account:

1. The whole text needs English polishing to substantially improve clarity;
2. The research question should be better defined.
3. Fig. 1 should bring more details and perform according to the reviewer’s suggestions.
4. In the discussion, the authors should bring a better interpretation of their data and how the work fill any gap in the literature revealing their novelty and/or importance in a more concise and clearer way to send the main message of the paper.
5. Incorporate some important literature about the topic that was not explored in the introduction and/or discussion.
6. Please see the attached file one of the reviewers sent and incorporate the suggestions made especially those regarding M&M and results.
7. Every other specific point highlighted by the reviewers should be carefully addressed.

Sincerely,

Gabriela Nardoto

Reviewer 1 ·

Basic reporting

I think this work is helpful for S. alterniflora manaement around the world. But some of problems should be clarified and revised before it could be published on PeerJ.

Experimental design

1, The research question should be further clarified. At now stage the research question is not defined.
2, The methods, i.e., the map of the waterlogging, mowing, tilling, should be illustrated with the in situ map, to show how the experiments were performed.

Validity of the findings

The author should provide the costs of each approach proposed in this research. Futhermore, it should be compared with other studies to show the novelty of their work.

Additional comments

It is an interesting work. I think it could be accepted if my concerns were reponded carefully.
1, The research question should be further clarified. At now stage the research question is not defined.
2, The methods, i.e., the map of the waterlogging, mowing, tilling, should be illustrated with the in situ map, to show how the experiments were performed.
3, The author should provide the costs of each approach proposed in this research. Futhermore, it should be compared with other studies to show the novelty of their work.
4, The author should provided a figure to show the spatial position, and the distribution of S. alterniflora at YRD.
5, The discussion part should be further improved to show the novelty of this work.

·

Basic reporting

Spartina alterniflora is a worldwide invasive species. Many countries and local governments are trying to control its invasion and restore native ecosystems. However, the controlling approaches largely depend on local situations. This work was preformed to control this species in in the Yellow River Delta, where is an important ecological zone. Authors successfully applied an integrated approach to control Spartina according to previous studies and gave suggestions on ecological restoration. The results have great reference significance for different places to carry out effective restoration. The paper was well wrote in English and provided clear figs/tabs. Here, I suggest authors to add another cases in other places in the Introduction section, such as Yangtze River estuary, and they can show a big picture of Spartina control.

Experimental design

In the Fig. 1, I suggest authors to add more information about experiment design, such as mechanical and chemical controls, timing, etc.

Validity of the findings

Validity of the findings is good.

Reviewer 3 ·

Basic reporting

The English of this article need to be revised by native speaker, because many of the sentences are too hard to understand.
The background of the study in the introduction part need to be added.

Experimental design

No comment.

Validity of the findings

No comment.

Additional comments

The article studied different approaches used to control S. alterniflora. This is useful for the protection and management of coastal wetland ecosystem. However, the artilce need to revised carefully, including the language and the structure or logic. For example, the "Material and method“ and "Results" should be revised to make them more concise. Specific comments are listed in the attacted file.

Annotated reviews are not available for download in order to protect the identity of reviewers who chose to remain anonymous.

---

## Round 0.2 · accepted · Accept

Dear authors,

the revised version of the manuscript is clearly an improved version of it that deserves now to be published in PeerJ. Thank you for addressing every reviewer's comments and reporting that clearly in your rebuttal letter. Congratulations!

Best regards,

Gabriela Nardoto

#